# Cross-Cultural Validation of the Healthcare Provider’s Practices, Attitudes, Self-Confidence, and Knowledge Regarding Bullying Questionnaire

**DOI:** 10.3390/healthcare12060606

**Published:** 2024-03-07

**Authors:** María del Carmen Celdrán-Navarro, Ismael Jiménez-Ruiz, A. Myriam Seva-Llor, James R. Moore, Cesar Leal-Costa

**Affiliations:** 1Servicio Murciano de Salud, 30120 Murcia, Spain; mariacarmen.celdran@um.es (M.d.C.C.-N.); jamesrichard.moore@um.es (J.R.M.); 2Faculty of Nursing, University of Murcia, El Palmar, 30120 Murcia, Spain; anamyriam.seva@um.es (A.M.S.-L.); cleal@um.es (C.L.-C.); 3ENFERAVANZA, Murcia Institute for BioHealth Research (IMIB-Arrixaca), El Palmar, 30120 Murcia, Spain

**Keywords:** validation, validation, bullying, primary healthcare provider

## Abstract

How can we know the reality of the context of bullying in the field of primary health care? The aim of this study is to obtain a validated and reliable tool that allows measurement of the involvement of primary care professionals in addressing bullying through a systematic content validation process. A cross-cultural validation of the Healthcare Provider’s Practices, Attitudes, Self-Confidence, and Knowledge Regarding Bullying Questionnaire was conducted for the Spanish perspective. This involved linguistic adaptation through translation–back-translation, content validity index (CVI) analysis, construct validity using confirmatory factor analysis (CFA), and internal consistency (Cronbach’s α). The total CVI was 0.95, with individual item scores ≥ 0.78. CFA revealed a good fit for the three subscales, with discrimination indices (item–total correlation within the dimension) > 0.30. Cronbach’s α for each dimension indicated a high level of reliability, with values of 0.735 for attitudes, 0.940 for self-confidence, and 0.895 for knowledge. The questionnaire is valid and reliable for evaluating the knowledge, attitudes, and self-confidence of primary care professionals in Spain regarding bullying. Its validity and reliability guarantee its potential use in other health settings and may lead to better training of professionals and school biopsychosocial health.

## 1. Introduction

Bullying is defined as a form of peer violence that occurs within the school context, characterized by repeated and intentional behavior that exploits an imbalance of power [1,2,3,4,5,6]. It can take various forms, generally classified into two types: direct (physical and verbal aggression such as kicks, punches, insults, nicknames, threats, sexual abuse, theft) and indirect (relational or social exclusion, blocking, manipulation) [1,2,5]. With the widespread use of technology of relationship, information, and communications (TRICs) in daily life, bullying has entered a new stage, leaving the walls of schools in the form of cyberbullying. Through internet-connected devices and social media platforms, cyberbullying allows for rapid, anonymous, and 24/7 dissemination of harmful content (intimidating or harmful material) [2,5].

Various roles are involved in this dynamic: actively, there are victims, aggressors, and those who play a dual role as victim–aggressors; passively, there are observers [4]. Observers can take reinforcing roles towards the aggressor or the victim. All may suffer the consequences of bullying, particularly those who are actively involved, leading to a deteriorated school climate. These repercussions can negatively impact the biopsychosocial wellbeing of those involved, causing short and long-term health disturbances. Physical effects include abdominal pain, headaches, back pain, loss of appetite, and sleep disorders. Mental consequences involving issues related to diminished self-esteem, stress, anxiety, depression, and aggression. Bullying can also have social implications such as school absenteeism, decreased sociolaboral resources (loneliness, poverty), and delinquent behaviors (alcohol and substance abuse, perpetuation of violent roles in future relationships). In severe cases, these repercussions may lead to fatal outcomes, including suicide [1,2,3,4,5,6].

Globally, according to United Nations Educational, Scientific and Cultural Organization (UNESCO), one-third of children experience bullying. In Spain, one in ten girls and adolescents will be victims of cyberbullying, a slightly higher percentage (12–15.4%) in the case of traditional bullying [7,8]. Mild or normalized forms of bullying, such as nicknames, insults, or minor physical aggression, are even more prevalent, with over half of all students (54%) experiencing them on a daily basis [9]. Given these statistics, bullying represents a globalized public health problem.

In light of the fact that bullying occurs both within and outside of schools and affects all social groups, interdisciplinary models must be implemented. Among various professional groups, primary healthcare, due to its competencies and knowledge of community health assets and influences on the family nucleus, should play an active part and leadership role [1,2,3]. Primary healthcare personnel, as the first point of contact with the healthcare system, have an essential role in the medical and school settings to prevent, detect, and manage the involvement of children and adolescents in episodes of violence [1]. Therefore, the guidelines from the National Institute for Health and Care Excellence (NICE) NG134, PH12, and PH20 emphasize the importance of the role of primary healthcare professionals, including the school nurse [10,11,12]. NICE guidelines stress that healthcare staff should be trained to enhance the assessment of psychosocial risk factors in the early stages, including anti-bullying strategies, as they have a position that encompasses the family and community perspective. They are knowledgeable about the potential involvement of all entities within it, to provide appropriate support and interventions for the child and their entire environment [12]. The guidelines include actions to provide a comprehensive program to develop social and emotional skills and improve the wellbeing of children. Recommendations include a curriculum that integrates social and emotional reflection across all subjects, addressing bullying prevention, teacher training to impart it effectively, and equipping them with tools for conflict resolution and addressing violent behaviors. It also involves supporting parents in parenting, promoting mental health [10,11,12].

The Program of Preventive Activities and Health Promotion [13] and the Health Education Program in Schools and Institutes of Spain [14], describe interventions at the three levels of prevention, addressing mental health issues and bullying behaviors in children and adolescents. They highlight the role of primary health care teams and specifically school nurses. They include actions that reject bullying, through democracy, respect for diversity, and the development of social skills, with a psychological approach and legal protection of minors [13,14].

Therefore, for primary healthcare professionals, especially school nurses, to carry out preventive and intervention work regarding bullying, it is necessary to have a favorable attitude towards addressing this issue and knowledge that provides them with intervention tools. To achieve this, there is a need to measure these aspects and conduct an analysis of the situation to adopt or improve clinical sessions and protocols focused on addressing school bullying [4].

In this regard, after having carried out a review of the most recent literature, the instrument developed by Hensley in 2015 was identified. The Healthcare Provider’s Practices, Attitudes, Self-Confidence, and Knowledge Regarding Bullying Questionnaire (HCP-PACK) [3], measures the actions of healthcare professionals in assisting children and adolescents involved in bullying, very similar to that pursued by this study. This tool consists of 63 items, divided into six blocks: sociodemographic, practices, attitudes, self-confidence, knowledge, and training needs. It yields three measurement subscales: attitudes, knowledge, and self-confidence. The response to each item is a four-point Likert scale: strongly agree, agree, disagree, and strongly disagree, assigned four, three, two, and one point/s, respectively. The psychometric properties were evaluated in three steps: content validation through expert feedback, reliability assessed through test–retest analysis with Pearson correlations, and finally, internal consistency of the three subscales using Cronbach’s alpha [3]. Therefore, it could be considered a useful tool in our context after translation and cross-cultural adaptation.

As reflected in the introduction of this article, well-established reasons justify and motivate the cross-cultural adaptation and validation of the measurement instrument. Questionnaires or tests are the most commonly used tools by researchers and professionals to acquire data on behavioral phenomena, especially those that significantly impact the lives of the individuals being observed or examined. Hence, it is essential that these instruments possess attributes of robustness and scientific quality [15]. Numerous recommendations have been established by various organizations, but how can we determine the reality of the context regarding school bullying within the primary healthcare setting? The aim of this study is to answer this question by obtaining a validated and reliable tool that allows us to measure the involvement of primary healthcare professionals in addressing school bullying through a systematic content validation process.

## 2. Materials and Methods

This is a transcultural adaptation and validation study of the Healthcare Provider’s Practices, Attitudes, Self-Confidence, and Knowledge Regarding Bullying Questionnaire (HCP-PACK) [3]. Authorization was obtained from the questionnaire’s author for the transcultural validation process. During the process, dimensions were eliminated, and items were adapted.

### 2.1. Description of the Original Instrument

The HCP-PACK questionnaire [3] is a tool designed to assess the capacity and willingness of healthcare professionals to address bullying in schools. It is divided into six areas: demographics, clinical practice, attitudes, knowledge, self-confidence, and training needs. Each area’s contents are as follows:Sociodemographic: Consisting of 6 questions about professional category, experience (years dedicated to working with children and whether it is in a good childcare setting), and work context.Clinical practice: Explores practices through 20 questions to identify cases of bullying among the population, including other adverse conditions such as lead toxicity and anemia. It also focuses on the reasons why participants decide to assess or not assess bullying.Attitudes: Addresses the facilities provided by the institution/service where professionals work regarding the care of the population experiencing bullying. It contains 6 questions with a value range from 6 to 24.Self-confidence: Assesses healthcare providers’ self-perception in assisting those involved in bullying. It includes 8 questions with scores ranging from 8 to 32.Knowledge: Assesses the knowledge that healthcare professionals have about bullying. Formed by 16 questions, with scores ranging from 16 to 64 points.Training needs: Asks 7 questions to healthcare providers about their opinion regarding the necessary training related to bullying and the most suitable format for it.

According to the author, this questionnaire is hetero-administered, with a response time of approximately 15 min, allowing quantification and comparison of healthcare providers’ responses through 63 items grouped into 3 subscales: attitudes, knowledge, and self-confidence. These are scored on a 4-point Likert scale: strongly agree (4 points), agree (3 points), disagree (2 points), and strongly disagree (1 point).

### 2.2. Translation and Cross-Cultural Adaptation

For the translation of the original version of the HCP-PACK questionnaire, a bilingual Spanish–English individual translated the English version into Spanish. The first version of the questionnaire (V1) was then developed. This version was back-translated by an independent bilingual Spanish–English person. Subsequently, it was retranslated into Spanish to generate Version 2. After verifying the grammatical, linguistic, and semantic correctness of this version, after which content validity was conducted. [16] For semantic and conceptual verification, a comparison was made between the original version and the translated version. In this process, the contributions of the experts were considered to make small changes to the items. Since it was a questionnaire carried out in a Western population, the adaptations were minor.

### 2.3. Content Validity

The translated Version 2 underwent expert judgment to evaluate the elements of the instrument and rate them based on their relevance and representativeness in the content domain [17]. As suggested by Enas Almanasreh et al., an expert panel of 5 to 10 members was formed. Specifically, the panel consisted of 9 experts, forming a multidisciplinary group with two main profiles: nursing and medicine. Six members were nurses, while the remaining three were physicians. Upon an in-depth analysis of their professional experience and training, it can be observed that they cover a broad spectrum of knowledge and perspectives.

Among the nurses, there were professionals specialized in different areas such as mental health, family and community care, and pediatrics. Additionally, with a research profile, including 5 doctors in nursing, one of whom has extensive experience in psychometrics.

In the group of physicians, the presence of specialists in pediatrics and family medicine is emphasized, with the latter being a medical doctor. Among the validation participants, it is noteworthy that the three physicians and two of the nurses work in the clinical field of primary care in close relation to the child and adolescent population. Moreover, one of the nurses providing care in hospital settings is in a mental health unit where children and adolescents involved in bullying are attended.

Content validity was analyzed, yielding an appropriate content validity index (CVI greater than or equal to 0.78) (see Table 1). However, due to the low scores of some items and the first of the four dimensions, the “Practices” dimension was eliminated because it obtained a CVI of 0.77. Additionally, item Auto3 from the self-confidence dimension and items C4, C5, and C9 from the knowledge dimension were reformulated. After generating Version 3 of the questionnaire, it was again subjected to expert panel evaluation, and content validity was analyzed, resulting in an appropriate content validity index (CVI) (CVI greater or equal to 0.78) [17] (see Table 2).

### 2.4. Procedure

The evaluation instrument was distributed among the nine previously described experts. The sample selection was convenience based, as specific profiles were sought to contribute to the questionnaire’s validity.

Distribution was carried out via email, containing a link that invited and directed participants to the survey, which they could complete and submit electronically. The email also provided explanations about the purpose of the survey, along with fundamental instructions for proper understanding. After the first phase, a second phase was conducted to identify discrepant scores. Following the second phase of CVI, the final version was distributed. It was sent as an online link via email, instant messaging groups, and social media, where the Instagram profile @acoso_escolar_enfermera was created to increase its reach. It was targeted towards primary care professionals: family and community care nurses, pediatric nurses, school nurses, family and community care physicians, pediatricians, and various medical and nursing internal residents in these specialties (Spanish acronyms MIR/EIR, respectively).

### 2.5. Data Analysis 

A descriptive analysis of the items was conducted (mean, standard deviation, skewness, and kurtosis). Item discrimination was calculated through corrected item–dimension correlation [16].

Subsequently, a confirmatory factor analysis (CFA) was performed to analyze the extent to which the scale items formed the established construct [18]. For CFA, the weighted least-squares mean and variance-adjusted (WLSMV) estimation method were used, suitable for ordered categorical data [19]. Data fit assessment to the models was performed using χ^2^/df, comparative fit index (CFI), Tucker–Lewis index (TLI), and root-mean-square error of approximation (RMSEA). Adequate fit is considered when χ^2^/df < 5, CFI > 0.90, TLI > 0.90, and RMSEA < 0.08 [20].

For item purification, descriptive statistics of items, discrimination indices, factor loadings of items in the dimension [21], and ensuring that the content of the dimensions was represented by the final items were used.

Reliability was analyzed as internal consistency with Cronbach’s alpha coefficient (α) for each dimension of the scale.

SPSS 22.0 [22] and Mplus 5.0 [19] were used for data analysis.

## 3. Results

A total of 299 responses were obtained from the questionnaire, of which 275 were refined for analysis. In 7 cases, consent was not provided, and in 17 cases, no records were filled, possibly due to a questionnaire access failure. The professional profile of the sample corresponds to the following: 40 family physicians, 23 pediatricians, 106 family and community nurses, 21 pediatric nurses, 24 school nurses, 47 medical and nursing residents, and 14 indicated other, including 13 generalist nurses and 1 general practitioner. The sample declared their gender, with 81.81% (*N* = 225) being female, compared to 18.18% (*N* = 50) male. Regarding years of work experience, the mean was 14.14 years (SD = 10.03 years) with a maximum of 45 years, a minimum of 1 year, and a mode of 4 years. Additionally, they were asked about their specialty; in this regard, 52.75% (*N* = 145) obtained their qualification through EIR/MIR, 12.72% (*N* = 35) through exceptional means, 2.18% (*N* = 6) through homologation, and finally, 32.36% (*N* = 89) did not possess any specialist title.

### 3.1. Item Analysis

Descriptive statistics for the items were obtained, revealing medium to high average scores for all items. The majority of items exhibited a normal distribution (skewness and kurtosis values within the range of 2 to −2), except for items A1, A2, A3, and A5, which had a leptokurtic distribution. The results of discrimination indices, measured through the corrected item–total correlation, were greater than 0.30 for all items (Table 3).

### 3.2. Internal Structure of the Scale: Confirmatory Factor Analysis (CFA) 

The final model with three oblique factors and 30 items was tested. The fit indices of the CFA showed χ^2^ = 1028.439 (df = 402; *p* < 0.001), CFI = 0.950, TLI = 0.946, and RMSEA = 0.076 (90% CI = 0.070−0.081). Factor loadings ranged from 0.153 for item 4 to 0.94 for item 2 (Figure 1). Based on these results, the fit of the data to the model is deemed appropriate.

### 3.3. Reliability Analysis

The internal consistency (Cronbach’s α) for each dimension of the scale was 0.735 for the attitude dimension, 0.940 for the self-confidence dimension, and 0.895 for the knowledge dimension.

## 4. Discussion

The main objective of this study was the cross-cultural validation of the original HCP-PACK questionnaire. Following this process, we can assert that the Spanish version of the instrument (hereafter referred to as HCP-PACKe) has resulted in a validated and reliable tool that allows us to measure the involvement and competencies of primary care professionals in addressing school bullying (View Appendix A and Appendix B). Consistent with Dale et al.’s research (2014), identifying the scope of the primary care team and measuring their competencies in addressing school bullying is crucial for providing comprehensive quality care to those involved in the bullying dynamics [1]. In the same vein, school nurses emphasize the need for multidisciplinary participation in improving the quality of interventions addressing this phenomenon [23].

Regarding the validation process, Hensley’s original questionnaire [3]. Measures various dimensions: clinical practice, attitudes, self-confidence, and knowledge of the primary care team in addressing school bullying. In response to our goal, this study translates and culturally validates the HCP-PACK into the Spanish context and language. During this process, the decision was made to retain only the dimensions of attitudes, self-confidence, and knowledge. Compared to the original questionnaire, this reduces the number of response items from 42 to 30 and eliminates the clinical practice dimension. In our study, this dimension from the original questionnaire is addressed as socio-labor variables. This decision was made to treat them as independent variables, allowing for a more detailed understanding of their individual influence on the studied phenomenon and the identification of possible causal relationships. Recent studies employing similar methodologies have also required the adaptation of dimensions and items to validate instruments related to expanding the practice of school nurses [24] and knowledge and attitudes regarding cutaneous reservoir management in primary care [25].

Our findings, as indicated by the factor analysis, show that the 30 items in the final Spanish version of the instrument have high internal consistency, with all items yielding results greater than 0.30. There is demonstrated linguistic equivalence with the original English version and comprehension of the final version based on the translation–back-translation process. Comparing reliability (internal consistency) assessed through Cronbach’s α coefficient in the three dimensions (attitudes, self-confidence, and knowledge) it is evident that both the original and Spanish questionnaires show similar results. In fact, after cross-cultural validation, greater stability is observed. Specifically, the results of said dimension are as follows: for attitudes, the original received 0.70, while the Spanish version achieved 0.735; in self-confidence, the initial version recorded 0.88, compared to 0.940; finally, in knowledge, the English version showed 0.84, while the Spanish version obtained 0.895 [3].

### 4.1. Implications for Primary Care Professionals’ Practice

Primary care serves as the starting point for coordinating and interconnecting resources for the health of users, especially those involved in bullying dynamics [1,4,5,6,12]. Assessing the quality of care and identifying areas for improvement in the care of those affected by school bullying requires measuring attitudes, knowledge, and actions implemented for this purpose [1,3,4,6,12]. The HCP-PACKe, being a valid and reliable tool, allows for the examination of care provided by primary care professionals in addressing the phenomenon of school bullying, enabling an analysis that differentiates between various professional categories. Future studies should focus on other healthcare settings to broaden the validity and reliability characteristics. Furthermore, with this tool, longitudinal studies could be carried out on knowledge, attitudes, and self-confidence regarding bullying; that would be susceptible to being related to the psychosocial wellbeing of the educational community. In addition, continuous adjustments can be investigated in the training and knowledge needs of health professionals, leading to continuous improvement in the quality of care and patient safety [3,4,26].

### 4.2. Limitations

There was an uneven representation among different professional groups, with nurses being the majority. This may lead to a lack of evidence on certain outcomes that could be obtained with a more proportionally sampled distribution of professional categories among participants. Furthermore, as stated in other research, convenience sampling and self-reporting can influence the results, as there may be a lack of honesty influenced by subjectivity and social acceptability [27,28].

Another significant factor that may limit results is the occasional nature of primary care staff, particularly nurses, who, due to their lack of continuity, may not be able to engage in school bullying dynamics or follow up on cases in the medium to long term.

Lastly, it is essential to consider that data were obtained in the recent post-pandemic era, during which, like most services, primary care teams rewrote their roles, leading to physically and psychologically exhausted teams [29,30].

## 5. Conclusions

The Spanish version of the HCP-PACK questionnaire has been developed based on cross-cultural validation methodology through IVC and CFA. Psychometric results indicate that it has resulted in a reliable and valid tool. The HCP-PACKe could provide primary healthcare professionals with an instrument to measure the involvement, confidence, and knowledge of primary care professionals in addressing bullying and the effect on individuals. This information is valuable for understanding how assistance is being provided in terms of prevention, treatment, and rehabilitation of those involved, as well as the potential needs and training methods perceived by healthcare professionals in this regard.

## Figures and Tables

**Figure 1 healthcare-12-00606-f001:**
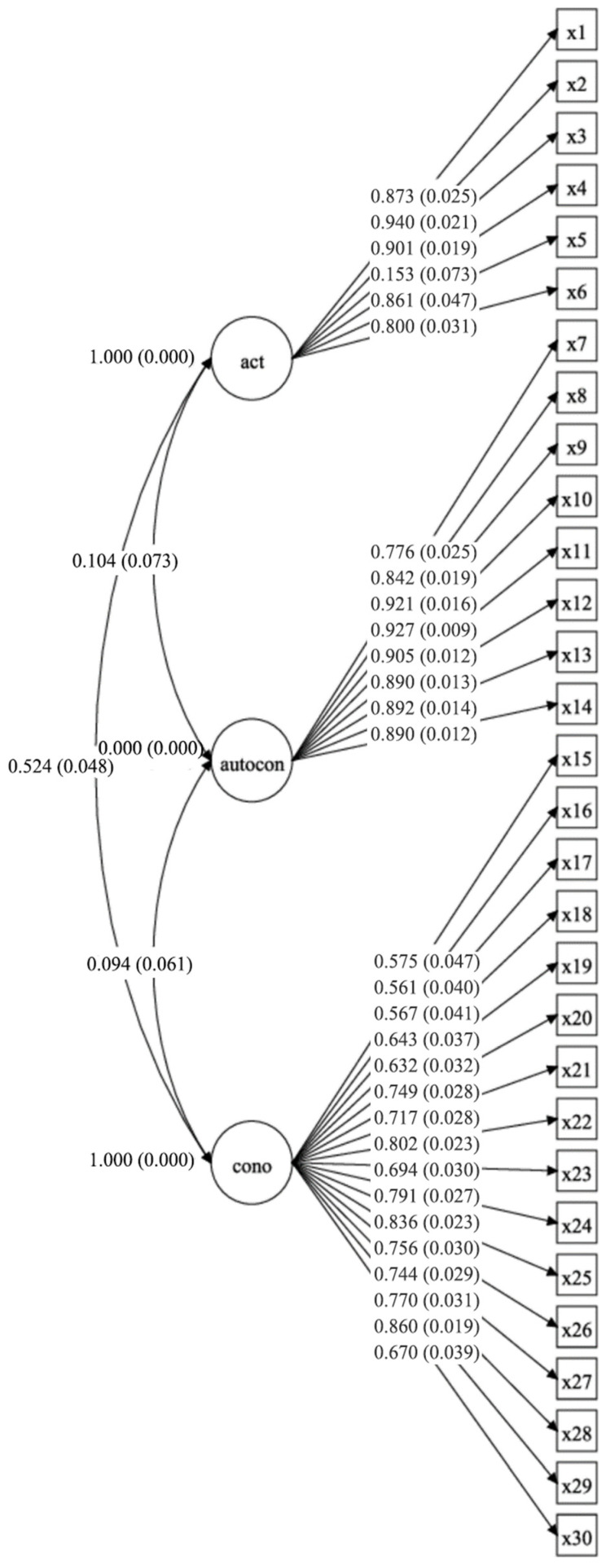
Confirmatory factor analysis of the Healthcare Provider’s Practices, Attitudes, Self-Confidence, and Knowledge Regarding Bullying Questionnaire (HCP-PACK).

**Table 1 healthcare-12-00606-t001:** Calculation of the CVI of the Healthcare Provider’s Practices, Attitudes, Self-Confidence, and Knowledge Regarding Bullying Questionnaire with the evaluations of nine experts. Phase 1.

Ítem	E1	E2	E3	E4	E5	E6	E7	E8	E9	CVI	No. of Agreements (3–4)
AC 1	4	3	4	4	4	4	3	4	4	1.00	9
AC 2	4	4	4	3	2	4	4	4	4	1.00	8
AC 3	4	4	3	4	2	4	4	4	4	0.89	8
AC 4	4	3	3	3	3	4	1	3	3	0.89	8
AC 5	4	4	4	4	3	3	4	3	4	1.00	9
AC 6	4	4	4	3	3	4	4	3	4	1.00	9
Auto 1	4	2	4	3	3	4	3	3	4	0.89	8
Auto 2	4	3	4	3	4	4	4	3	4	1.00	9
Auto 3	4	2	4	2	2	4	3	3	4	0.67	6
Auto 4	4	3	4	4	3	3	3	4	3	1.00	9
Auto 5	4	2	4	4	3	4	3	3	4	1.00	9
Auto 6	3	2	3	3	2	4	3	3	4	0.78	7
Auto 7	3	2	3	4	3	4	3	3	3	0.89	8
Auto 8	4	2	3	4	2	4	4	3	4	0.78	7
C1	4	4	3	3	3	4	4	3	3	1.00	9
C2	4	4	3	2	3	4	4	3	3	0.89	8
C3	4	3	3	3	2	4	3	3	2	0.78	7
C4	4	3	2	2	2	4	4	4	2	0.56	5
C5	4	2	3	4	2	4	4	2	3	0.67	6
C6	4	4	4	4	3	4	4	3	4	1.00	9
C7	4	4	3	4	3	4	4	2	4	0.89	8
C8	4	4	3	3	3	4	4	3	3	1.00	9
C9	4	4	2	3	2	4	4	2	3	0.67	6
C10	4	4	3	3	3	4	4	3	4	1.00	9
C11	3	4	3	4	3	4	4	3	4	1.00	9
C12	3	4	4	2	2	4	4	3	4	0.78	7
C13	4	4	4	3	3	4	4	4	4	1.00	9
C14	4	4	4	4	3	4	4	4	3	1.00	1
C15	4	4	4	3	3	4	4	4	4	1.00	9
C16	4	4	3	4	3	4	3	4	4	1.00	9
TOTAL CVI										0.90	
Response scale 1 = not relevant; 2 = something relevant; 3 = quite relevant; 4 = very relevant											

CVI, content validity index. AC = Attitude dimension, Auto = Self-confidence dimension, C = Knowledge dimension.

**Table 2 healthcare-12-00606-t002:** Calculation of the CVI of the Healthcare Provider’s Practices, Attitudes, Self-Confidence, and Knowledge Regarding Bullying Questionnaire with the evaluations of nine experts. Phase 2.

Ítem	E1	E2	E3	E4	E5	E6	E7	E8	E9	CVI	No. of Agreements (3–4)
AC 1	4	4	4	4	4	4	3	3	4	1.00	9
AC 2	4	4	3	4	4	3	4	3	4	1.00	9
AC 3	3	4	4	4	4	4	3	3	4	1.00	9
AC 4	3	3	4	2	4	3	3	2	3	0.89	7
AC 5	4	4	3	4	4	4	4	3	4	1.00	9
AC 6	3	4	4	4	4	4	3	3	4	1.00	9
Auto 1	4	4	4	3	4	4	3	3	3	1.00	9
Auto 2	4	4	4	4	4	4	4	3	3	1.00	9
Auto 3	4	4	2	3	4	3	3	3	2	0.78	7
Auto 4	3	4	4	4	4	4	4	3	3	1.00	9
Auto 5	3	4	4	3	4	4	3	3	3	1.00	9
Auto 6	3	3	4	4	4	3	3	3	3	1.00	9
Auto 7	3	4	4	3	4	4	4	3	2	0.89	8
Auto 8	3	4	4	3	4	3	3	3	3	1.00	9
C1	3	4	3	4	4	4	4	4	4	1.00	9
C2	3	3	4	4	4	4	3	3	4	1.00	9
C3	3	3	3	3	4	3	3	2	1	0.78	7
C4	3	3	4	4	3	3	3	2	3	0.89	8
C5	2	4	3	4	4	4	3	2	4	0.89	7
C6	3	4	3	4	4	4	4	3	4	1.00	9
C7	2	4	4	4	4	4	4	2	4	0.78	7
C8	3	4	4	4	4	4	4	3	4	1.00	9
C9	2	3	4	4	4	3	3	3	4	0.89	8
C10	3	4	4	4	4	4	4	4	4	1.00	9
C11	3	4	3	4	4	4	4	2	4	0.89	8
C12	3	4	3	4	4	4	3	3	4	1.00	9
C13	4	4	4	4	4	4	4	3	4	1.00	9
C14	3	4	4	4	4	4	4	3	4	1.00	1
C15	4	4	4	4	4	4	3	3	4	1.00	9
C16	3	4	3	3	3	4	3	3	4	1.00	9
TOTAL CVI										0.95	
Response scale 1 = not relevant; 2 = something relevant; 3 = quite relevant; 4 = very relevant											

CVI, content validity index AC = Attitude dimension, Auto = Self-confidence dimension, C = Knowledge dimension.

**Table 3 healthcare-12-00606-t003:** Descriptive Statistics and item–total dimension correlation.

Ítem	Min	Max	M	SD	Skewness	Kurtosis	Item–Total Dimension Correlation
AC1	1	4	3.57	0.66	−1.73	3.40	0.61
2. AC2	1	4	3.76	0.55	−2.71	8.61	0.69
3. AC3	1	4	3.70	0.61	−2.47	6.94	0.70
4. AC4	1	4	2.56	1.15	0.01	−1.45	0.17
5. AC5	1	4	3.83	0.48	−3.64	16.18	0.46
6. AC6	1	4	3.53	0.71	−1.59	2.44	0.62
7. Auto1	1	4	2.49	0.69	−0.21	−0.22	0.67
8. Auto2	1	4	2.30	0.76	0.24	−0.22	0.76
9. Auto3	1	4	2.23	0.72	0.49	0.29	0.81
10. Auto4	1	4	2.24	0.76	0.42	0.03	0.84
11. Auto5	1	4	2.22	0.74	0.28	−0.10	0.82
12. Auto6	1	4	2.24	0.73	0.26	−0.09	0.80
13. Auto7	1	4	2.16	0.72	0.58	0.54	0.81
14. Auto8	1	4	2.16	0.71	0.44	0.32	0.79
15. C1	1	4	3.65	0.52	−1.25	1.50	0.39
16. C2	1	4	3.05	0.83	−0.45	−0.60	0.48
17. C3	1	4	2.50	0.84	0.11	−0.55	0.48
18. C4	1	4	2.62	0.87	0.25	−0.86	0.59
19. C5	1	4	2.87	0.83	−0.26	−0.60	0.58
20. C6	1	4	3.39	0.55	−0.27	−0.03	0.61
21. C7	1	4	3.05	0.75	−0.45	−0.11	0.63
22. C8	1	4	3.55	0.58	−1.13	1.50	0.64
23. C9	1	4	2.92	0.81	−0.32	−0.46	0.61
24. C10	2	4	3.41	0.54	−0.13	−1.03	0.65
25. C11	2	4	3.48	0.54	−0.27	−1.17	0.64
26. C12	2	4	3.40	0.58	−0.33	−0.74	0.60
27. C13	2	4	3.56	0.51	−0.40	−1.45	0.52
28. C14	2	4	3.43	0.58	−0.41	−0.74	0.61
29. C15	1	4	3.56	0.55	−1.03	1.57	0.65
30. C16	1	4	3.47	0.68	−1.35	2.11	0.46

M: Mean; SD: standard deviation AC = Attitude dimension, Auto = Self-confidence dimension, C = Knowledge dimension.

## Data Availability

The data are available upon email request to the corresponding authors.

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
