# Peer review of "Cross-Cultural Validation of the Healthcare Provider’s Practices, Attitudes, Self-Confidence, and Knowledge Regarding Bullying Questionnaire"

_healthcare, 2024, doi:10.3390/healthcare12060606_

Round 1
Reviewer 1 Report
Comments and Suggestions for Authors
Dear Authors,
Thank you for the opportunity to review this paper entitled "Cross-cultural validation of the Healthcare Provider’s Practices, Attitudes, Self-confidence, & Knowledge Regarding Bullying Questionnaire." It is an important subject, and good to research it. Some points need to be revised, and my suggestions are as follows:
Introduction
1. Page 1, Line 36: What is the meaning of “24/7 dissemination of harmful content”?
2. Page 2, Line 50: The abbreviation “UNESCO” is first time mentioned; write the full name.
3. The literature review lacks past research on types of bullying questionnaires.
4. What is the usage status of The Healthcare Provider’s Practices, Attitudes, Self-confidence, & Knowledge Regarding Bullying Questionnaire (HCP-PACK)? Why is it suitable for this study? The reason for deleting the other three variables(sociodemographic, practices, and training needs) is not specified.
Methodology
- Page 7, Line 223: What is the EIR/MIR ?
- The explanation of why the scale was reduced to three concepts is not specific enough. In particular, clinical experience and sociodemographic are important influencing factors.
- The author said that this is a cross-cultural validation methodology, but it only presents the two-way translation process of a general questionnaire and does not clearly describe what modifications are due to cultural factors. It is recommended that the process should be clearly explained.
- Research Tools: Give examples to illustrate the topic content of each variable.
Result
Page 7, Line 211: Mplus 5.0[17] was used for data analysis.==> I'm not familiar with this statistical method. The statistical results must be reviewed by statistical experts.
References
Some references have the wrong format.
Author Response
Reviewers 1's comments:
Dear Reviewer:
We sincerely appreciate your constructive comments on our article. We have taken each of your points into consideration and have made the following modifications:
Comment 1. Page 1, Line 36: What is the meaning of “24/7 dissemination of harmful content”?
- Response:
In the context of cyberbullying, it refers to the ability to spread intimidating or harmful material over the Internet at any time of the day. This means that content can be shared, replicated and even modified to increase its negative impact without time restrictions, as online platforms are constantly available and running. This relentless accessibility through internet-connected devices expands the potential of cyberbullying, allowing bullying to persist without interruption and have a much broader reach than traditional bullying. Explanation is included in parentheses on line 36-37.
Comment 2. Page 2, Line 50: The abbreviation “UNESCO” is first time mentioned; write the full name.
- Response:
United Nations Educational, Scientific and Cultural Organization (UNESCO), included in text line 51.
Comment 3. The literature review lacks past research on types of bullying questionnaires.
- Response:
Clarified on page 2, lines 79-84
Comment 4. What is the usage status of The Healthcare Provider’s Practices, Attitudes, Self-confidence, & Knowledge Regarding Bullying Questionnaire (HCP-PACK)? Why is it suitable for this study? The reason for deleting the other three variables (sociodemographic, practices, and training needs) is not specified.
- Response:
The original questionnaire is composed of 5 dimensions: clinical practice, attitudes, knowledge, self-confidence and training needs. The dimensions “clinical practice”, which describes the interventions carried out on bullying and “training needs”, which describes the training and training needs on bullying, were excluded from the construction of the instrument because they did not address behavioral phenomena. Clinical practice and training needs, although they are not part of the dimensions of the new questionnaire, will be included within the socio-demographic and professional variables. In this way, the socio-demographic variables, training needs and the description of clinical practice are treated as independent variables, thus allowing a more detailed understanding of their individual influence on the phenomenon studied and the identification of possible causal relationships. The behavioral dimensions: attitudes, knowledge and self-confidence will be the dimensions that the new questionnaire would measure.
Comment 5. Page 7, Line 223: What is the EIR/MIR ?
- Response: EIR Spanish acronym for nursing internal residents, MIR Spanish acronym for medical internal residents, page 5 line 204
Comment 6. The explanation of why the scale was reduced to three concepts is not specific enough. In particular, clinical experience and sociodemographic are important influencing factors.
- Response:
Indeed, sociodemographic variables and clinical experience are factors that can influence the phenomenon, but as we have commented in comment 4, they will be treated as independent variables and not as a dimension of the questionnaire. Likewise, both dimensions have non-unified item measurement systems. For example: the “clinical practice” dimension of the original questionnaire includes items with diverse scoring systems: some items include dichotomous yes/no responses, other items include a 4-point Likert scale, and other items include 5-point Likert scales. This breaks the uniformity of the measurements made.
Comment 7. The author said that this is a cross-cultural validation methodology, but it only presents the two-way translation process of a general questionnaire and does not clearly describe what modifications are due to cultural factors. It is recommended that the process should be clearly explained.
- Response:
The origin of the original questionnaire is based on a western style culture, so the cultural differences with the country where the validation will be applied (Spain) are not substantial. Likewise, during the content validation process and in order to carry out a semantic and conceptual verification, a comparison was made between the original version and the translated version. In this process, the contributions of the experts were considered to make small changes to the items. Since it was a questionnaire carried out within a Western population, the adaptations were minor. Information included on lines 160-161.
Comment 8. Research Tools: Give examples to illustrate the topic content of each variable.
- Response:
Examples of the items are explained in Appendix A.
Commet 9. Some references have the wrong format.
- Response: All references have been reviewed.

Reviewer 2 Report
Comments and Suggestions for Authors
Thank you for conducting such a cross-cultural validation tool. There are few notes that can enhance the quality of the paper.
1. Abstract: need more elaboration of the implication of the results in practice.
2. Introduction:
a) Follow the Journal format in regard to citation. The number should be in parenthesis then a period.
b) Line 33, communications not communication.
c) line 36, numbers 2, and 5 should be in the Journal format.
d) In the introduction, the reader wants to know a little bit about the Spanish culture in regard to bullying.
e) It is confusing for the reader to grasp from the beginning that you didnot include the clinical practice subscale in your study until a while. Also, the title gives sense that you also measure the practices. therefore, this could be improved.
f) It is confusing that you used self-confidence and self-efficacy interchangeably.
Materials and methods
a) Are there any scoring system for clinical practice and training needs subscales in line 121 and 133, respectively.
b) Do you have any reference for using the procedure of translation and backtranslation (in lines 141-143).
c) it is not clear how you approached and invited the participants (in lines 185-194).
Results
a )what are the EIR/MIR in lines 223?
b) Would you describe how did you reach to the cut-off points regarding the medium to high average scores in line 226?
Discussion
a) could you explain and discuss how the cross-validation between your culture and the original tool cultures' was assured. Is there any similarities or differences in cultures?
b) do you think using a convenience sampling technique and self-report may limit the generalization of your results?
c) Suggest any future research that can be done in this area.
References:
a) follow the format of the journal
B) see number 3 in line 338 for caps.
Comments on the Quality of English Language
it needs minor English revision. There are few long sentences. Few typological issues. There are also paragraphs with single sentence.
Author Response
Reviewers 2's comments:
Dear Reviewer:
We sincerely appreciate your constructive comments on our article. We have taken each of your points into consideration and have made the following modifications:
We have considered your suggestions and made the following changes:
Comment 1. Abstract: need more elaboration of the implication of the results in practice.
- Response:
Included new wording on lines 21-24.
Comment 2. Introduction. Follow the Journal format in regard to citation. The number should be in parenthesis then a period.
- Response:
Done
Comment 3. Introduction. Line 33, communications not communication.
- Response:
Done
Comment 4. Introduction. line 36, numbers 2, and 5 should be in the Journal format.
- Response:
Done
Comment 5. In the introduction, the reader wants to know a little bit about the Spanish culture in regard to bullying.
- Response:
Included in text, page 2 lines 79 - 85
Comment 6. Introduction It is confusing for the reader to grasp from the beginning that you didnot include the clinical practice subscale in your study until a while. Also, the title gives sense that you also measure the practices. therefore, this could be improved.
- Response:
The original questionnaire is composed of 5 dimensions: clinical practice, attitudes, knowledge, self-confidence and training needs. The dimensions “clinical practice”, which describes the interventions carried out on bullying and “training needs”, which describes the training and training needs on bullying, were excluded from the construction of the instrument because they did not address behavioral phenomena. Clinical practice and training needs, although they are not part of the dimensions of the new questionnaire, will be included within the socio-demographic and professional variables. In this way, the socio-demographic variables, training needs and the description of clinical practice are treated as independent variables, thus allowing a more detailed understanding of their individual influence on the phenomenon studied and the identification of possible causal relationships. The behavioral dimensions: attitudes, knowledge and self-confidence will be the dimensions that the new questionnaire would measure. It should be noted that the “clinical practice” dimension of the original questionnaire does not function as a dimension in itself, it is various independent items that include professional information. Likewise, both dimensions have non-unified item measurement systems.
Comment 7. It is confusing that you used self-confidence and self-efficacy interchangeably.
- Response:
I have corrected this in the text: page 2: line 99, page: 4 line 139.
Comment 8. Materials and methods Are there any scoring system for clinical practice and training needs subscales in line 121 and 133, respectively.
- Response:
The “clinical practice” dimension of the original questionnaire includes items with diverse scoring systems: some items include dichotomous yes/no responses, other items include a 4-point Likert scale, and other items include 5-point Likert scales. This breaks the uniformity of the measurements made.
Comment 9. Materials and methods Do you have any reference for using the procedure of translation and backtranslation (in lines 141-143).
- Response:
Added reference.
Comment 10. Materials and methods it is not clear how you approached and invited the participants (in lines 185-194).
- Response:
As indicated in the manuscript, page 5 lines 183-189, the final version was sent as an online link through email, instant messaging groups and social networks, where the Instagram profile @acoso_escolar_enfermera was created to increase its reach. It was aimed at primary care professionals: family and community care nurses, pediatric nurses, school nurses, family and community care physicians, pediatricians, and various medical and nursing interns in these specialties; (Acronyms MIR/EIR, respectively).
Comment 11. Results. what are the EIR/MIR in lines 223?
- Response:
Clarified in the text page 5, line 204
Comment 12. Results. what are the EIR/MIR in lines 223?
- Response:
Clarified in the text page 5, line 204
Comment 13. Discussion. Could you explain and discuss how the cross-validation between your culture and the original tool cultures' was assured. Is there any similarities or differences in cultures?
- Response:
The origin of the original questionnaire is Western, so the cultural differences with the country where the validation will be applied (Spain) are not substantial. Likewise, during the content validation process and in order to carry out a semantic and conceptual verification, a comparison was made between the original version and the translated version. In this process, the contributions of the experts were considered to make small changes to the items. Since it was a questionnaire carried out in a Western population, the adaptations were minor. Information included on lines 146-150.
Comment 14. Discussion. Do you think using a convenience sampling technique and self-report may limit the generalization of your results?
- Response:.
I agree with your opinion, I add it to the manuscript, on page 7 lines 309 -311.
Comment 15. Discussion. c) Suggest any future research that can be done in this area.
- Response:
They are indicated in the Implications for Primary Care Professionals' Practice section. Even so, agreeing with you, I expand the information in said section, page 7 lines 299, 304
Comment 16. References. follow the format of the journal
- Response:
All references have been reviewed.
Comment 17. References. see number 3 in line 338 for caps.
- Response:
Done

Round 2
Reviewer 1 Report
Comments and Suggestions for Authors
The revised manuscript is more clear and can respond to questions.